# *PYL* Family Genes from *Liriodendron chinense* Positively Respond to Multiple Stresses

**DOI:** 10.3390/plants12142609

**Published:** 2023-07-11

**Authors:** Xinru Wu, Junjie Zhu, Xinying Chen, Jiaji Zhang, Lu Lu, Zhaodong Hao, Jisen Shi, Jinhui Chen

**Affiliations:** 1State Key Laboratory of Tree Genetics and Breeding, Co-Innovation Center for Sustainable Forestry in Southern China, Nanjing Forestry University, Nanjing 210037, China; wxr0410@outlook.com (X.W.); zhujunjie19981009@njfu.edu.cn (J.Z.); chenxinying0531@163.com (X.C.); zhangjiaji0609@163.com (J.Z.); lulu2020@njfu.edu.cn (L.L.); haozd@njfu.edu.cn (Z.H.); jshi@njfu.edu.cn (J.S.); 2Key Laboratory of Forest Genetics and Biotechnology of Ministry of Education, Nanjing Forestry University, Nanjing 210037, China; 3College of Landscape Architecture, Nanjing Forestry University, Nanjing 210037, China

**Keywords:** *LcPYL*, ABA signaling, abiotic and biotic stresses, *Liriodendron hybrid*

## Abstract

The phytohormone abscisic acid (ABA) plays important roles in response to abiotic and biotic stresses in plants. Pyrabactin resistance 1-like (PYR/PYL) proteins are well-known as ABA receptors, which are responsible for ABA signal transduction. Nevertheless, the characteristics of *PYL* genes from *Liriodendron chinense*, an endangered timber tree, remain unclear in coping with various stresses. In this study, five *PYL*s were identified from the genome of *Liriodendron chinense* by sequence alignment and conserved motif analysis, which revealed that these *LcPYL*s contain a conserved gate and latch motif for ABA binding. The LcPYL promoters possess a series of cis-acting elements involved in response to various hormone and abiotic stresses. Moreover, the transcriptome data of *Liriodendron hybrid* leaves reveal that *LcPYL* genes specifically transcript under different abiotic stresses; *Lchi11622* transcription was induced by drought and cold treatment, and *Lchi01385* and *Lchi16997* transcription was upregulated under cold and hot stress, respectively. Meanwhile, the *LcPYL*s with high expression levels shown in the transcriptomes were also found to be upregulated in whole plants treated with the same stresses tested by qPCR. Moreover, under biotic stress caused by scale insect and whitefly, *Liriodendron hybrid* leaves exhibited a distinct phenotype including disease spots that are dark green in the middle and yellow on the margin; the qPCR results showed that the relative expression levels of *Lchi13641* and *Lchi11622* in infected leaves were upregulated by 1.76 and 3.75 folds relative to normal leaves, respectively. The subcellular localizations of these stress-responsive LcPYLs were also identified in protoplasts of *Liriodendron hybrid*. These results provide a foundation to elucidate the function of *PYL*s from this elite tree species and assist in understanding the molecular mechanism of *Liriodendron hybrid* in dealing with abiotic and biotic stresses. In future research, the detailed biological function of *LcPYL*s and the genetic redundancy between *LcPYL*s can be explored by gene overexpression and knockout based on this study.

## 1. Introduction

Higher plants have evolved a high plasticity for adaptation to environmental challenges such as drought, cold, hot and insect attack stresses, which severely influence normal plant development and productivity [1,2,3,4]. Phytohormone abscisic acid (ABA) plays a critical role in plant growth and development, including cell division and elongation, embryo maturation, seed dormancy and germination, leaf senescence, root growth, fruit ripening and stomatal aperture [5,6]. ABA also serves as an endogenous messenger to transduce the signals of various stresses and plays a major role in plant adaptation to the environment [7,8,9]. For instance, in response to drought stress, plants synthesize the hormone ABA to reduce water loss by triggering stomata closure [10]. However, although rapid production of ABA in response to environmental stresses is necessary to define ABA as a stress hormone, decoding ABA signaling is also essential in that role. Therefore, the perception and integration of environmental stresses induced by ABA are considered as important research interests for improving plant performance [11].

ABA receptor proteins consist of the pyrabactin resistance1 (PYR1) and PYR1-like (PYL) proteins [12]. Following a series of crystallographic studies, a well-accepted mechanism of ABA recognition by PYLs was proposed. ABA signaling is perceived and transduced by a gate–latch–lock mechanism; the binding of ABA induces conformational changes in the PYL receptor ligand binding pocket. This creates an interaction surface for downstream clade A type 2C protein phosphatases (PP2Cs), leading to the release of class III sucrose non-fermenting 1-related protein kinase 2s (SnRK2s) [13,14,15,16]. Free and activated SnRK2s phosphorylate the downstream targets such as ABA-responsive elements (ABREs)/ABA binding factors (ABFs) [17,18]. This unique ABA perception mechanism reveals the importance of *PYL*s in the ABA signaling transduction module. Therefore, numerous studies have identified *PYR*/*PYL*s from various plant species. A total of 14 *AtPYR*/*AtPYL*s were verified from model plant *Arabidopsis* [19]. A total of 9 *PYL*s from grape [20], 12 *PYL*s from rice [21], 14 *PYL*s from tomato [22], 14 *PYL*s from rubber tree [23], 14 *PYL*s from *Camellia sinensis* [24], 21 *PYL* homologs from soybean [25], and 27 *PYL*s from cotton [26] have already been identified by researchers. However, the function of most *PYL*s is still a conundrum, which might be caused by genetic redundancy in ABA signaling [19].

The *Liriodendron* genus belongs to the magnolia family, containing one pair of sister species: *Liriodendron tulipifera* (*L. tulipifera*) and *Liriodendron chinense* (*L. chinense*) [27,28]. *Liriodendron hybrid* (*L. hybrid*) was obtained via the cross between *L. tulipifera* and *L. chinense*, which is a type of ornamental tree used for landscaping because of its straight trunk, distinctive leaf shape and tulip-shaped flowers. *L. hybrid* plants are also widely cultivated as industrial timber trees due to their versatile wood with excellent working properties. Furthermore, *L. hybrid* can survive in areas with harsh environments, such as drought mountains with cold temperature. To investigate the mechanisms of *L. hybrid* in coping with environmental stresses, in this study, we identified *PYL*s from the *L. chinense* genome (*LcPYL*s) and studied their response to drought, extreme temperature and insect infection. The results of sequence alignments and conserved domain analysis further verified the five isolated candidates as PYL members. The evolutionary study of LcPYLs and their homologous proteins from other plants showed a consistent classification with phylogenetic analysis of AtPYLs. Moreover, the identified *LcPYL*s were found to specifically respond to drought, hot, cold, and insect attack stresses. Our results provide evidence for the response of *L. hybrid* ABA receptors under various stresses, providing a foundation for functional exploration of *LcPYL* genes and better understanding the mechanism of stress tolerance in this relict tree species.

## 2. Results

### 2.1. Genome-Wide Analysis and Chromosome Distribution of LcPYL Family Genes

To explore the molecular mechanism of the ABA signaling pathway in *L. hybrid* under abiotic and biotic stresses, we identified the ABA receptors from the genome of *L. chinense*. As a result, 13 *LcPYL* genes were obtained by searching with the conserved domains, including Polyketide_cyc2 domain (PF10604) and Polyketide_cyc domain (PF03364) [16]. To further verify the *LcPYL* members, we used the NCBI-CDD database to identify whether the candidates have a PYR/PYL (RCAR)-like domain (cd07821). As a consequence, six *PYL*s were identified from the *L. chinense* genome. Combined with ABA binding conserved motif analysis, we finally obtained five *PYL* genes from the *L. chinense* genome.

The ORF sizes of these PYLs vary from 534 bp to 648 bp (Figure 1A). Analysis of the physical and chemical properties of LcPYL proteins showed that Lchi16997 has the smallest protein sequence (197 aa), followed by Lchi11622 (178 aa), and Lchi01385 has the largest number of amino acids (204 aa) (Table 1). The molecular weights of identified proteins ranged from 19.75 KDa to 23.75 KDa. The range of isoelectric points (pI) is from 5 to 7.65. Additionally, the predicted subcellular locations of LcPYL proteins showed that Lchi16997 and Lchi01385 are located in the cytoplasm, whereas Lchi13641, Lchi00864 and Lchi11622 are located in chloroplast (Table 1). The chromosomal distributions of LcPYLs showed that five PYLs were located in five chromosomes of L. chinense (19 chromosomes in total), and Lchi00864, Lchi11622, Lchi13641, Lchi01385 and Lchi16997 were distributed on chromosomes 1, 3, 5, 11 and 15 of L. chinense, respectively (Figure 1B).

### 2.2. Conserved Domains and Phylogenetic Study of LcPYL Proteins

To confirm the conserved sequences, we compared the protein sequences of all LcPYLs. The result revealed that although a high difference exists in the LcPYL protein sequences, the conserved motifs that are necessary for ABA receptors are included in all LcPYL proteins. A gate motif containing the SGPLA sequence and a latch motif containing the HRL sequence were found to be the common characteristics in all LcPYL proteins (16) (Figure 2A). The gate–latch–lock structure for ABA signaling transduction was also observed in the predicted 3D structures for each LcPYL protein (Figure 2B–F). The prediction of secondary structures showed that alpha helix and random coil are the most common secondary structures in all LcPYL proteins, followed by extended strand and beta turn (Table 2).

PYLs from *Arabidopsis thaliana* (*A. thaliana*, AtPYLs) have been well-studied both in terms of function and structure. Therefore, to further verify the candidate genes from *L. chinense* as *PYL* family members, we analyzed the phylogenetic relationship between LcPYLs and AtPYLs and explored the conserved motifs in two groups of PYLs. The phylogenetic study showed that 5 LcPYLs and 14 AtPYLs were classified as three clades in general (Figure 3A). Clade I includes Lchi01385, Lchi00864, AtPYR1 and AtPYL1–3; Lchi01385 is a sister clade with AtPYR1 and AtPYL1; and Lchi00864 is classified to the subgroup containing AtPYL2 and AtPYL3. Clade II contains Lchi13641, Lchi11622, AtPYL4–6 and AtPYL11–13; Lchi13641 is a sister branch with AtPYL4; and Lchi11622 is a sister branch of AtPYL11–13. Clade III contains Lchi16997 and AtPYL7–10; however, Lchi16997 is closer to AtPYL8 and AtPYL10 (Figure 3A). Although the result of conserved motif exploration by MEME software with 10 conserved motifs as the cutoff revealed the specific motifs in each group of PYLs, the numbers of all PYLs are no more than five. In addition, all PYLs harbor motifs 1–3, and motif 1 includes the gate and latch sequences for ABA binding (Figure 3B,C).

To better understand the phylogenetic relationship between LcPYLs and their homologous proteins, the amino acid sequences of PYLs from *A. thaliana*, *Oryza sativa* (*O. sativa* L.) and *Vitis vinifera* (*V. vinifera* L.) were used to construct a phylogenetic tree using TBtools (v1.108) and the maximum-likelihood method with 1000 bootstrap replications (Figure 4). Five LcPYLs, fourteen AtPYLs, twelve OsPYLs and nine VvPYLs were classified into three clades (clade I, clade II and clade III) based on the classification of AtPYLs [12] (Figure 4). Overall, clade I contains 11 PYL members, clade II contains 15 PYLs, and clade III contains 14 PYLs (Figure 4). These results further confirm that the candidates from *L. chinense* are *PYL* family genes.

### 2.3. Cis-Acting Element Exploration of LcPYL Promoters

The result of the phylogenetic study and conserved motif analysis revealed the common and unique motifs in different groups of PYLs that may indicate the specific functions of different PYLs. Therefore, to explore the potential function of *LcPYL*s, we analyzed the cis-acting elements of *LcPYL* promoters to understand the transcription regulation. To this end, 3000 bp upstream sequences of *LcPYL* ORFs (including 5′UTR) were extracted from the *L. chinense* genome to predict cis-acting elements using the PlantCARE database. The results show that the regulatory elements of the *LcPYL*s were abundant in number and variety. There were six hormone-related cis-acting elements from the prediction, including ABA, MeJA, gibberellin, salicylic acid (SA), ethylene and auxin. All five *LcPYL* promoters harbor at least two hormone response elements. ‘ABRE’ and ‘AAGAA’ identified as ABA response elements were found in all *LcPYL* promoters as expected (Figure 5A). Additionally, cis-acting elements involved in stress response were also found in *LcPYL* promoters. *Lchi11622*, *Lchi00864* and *Lchi01385* promoters harbor drought-inducible element ‘MBS’, whereas *Lchi00864* and *Lchi13641* promoters contain the cis-acting element involved in defense and stress-responsiveness ‘TC-rich repeats’ (Figure 5A). The statistics showed that all *LcPYL* promoters contain at least six kinds of cis-acting elements related to hormone and multiple stresses (Figure 5B). The diversities of cis-acting elements in the *LcPYL* promoters suggest that the transcription of these genes might be regulated by the corresponding hormone and stress factors.

### 2.4. Expression Pattern of LcPYLs in Different Tissues of L. hybrid

To analyze the expression pattern of *PYL* genes in different tissues of *L. hybrid*, previously published RNA-seq data of phloem, stigma, xylem, bud, stamen, leaf, bark and sepal were used for elucidation (Figure 6). The transcript abundance of *Lchi00864* in all harvested tissues was hardly detected. On the contrary, *Lchi16997* has high transcription in most of tissues, except phloem and xylem tissues. *Lchi13641* expresses in the sepal, bark, leaf, stamen and stigma but not in the phloem, xylem or bud. *Lchi11622* specifically transcripts in bark, bud and xylem, whereas *Lchi01385* specifically expresses in bark, leaf and stamen but with a relatively lower level than *Lchi13641* and *Lchi16997* (Figure 6). These results indicate the potential function of the target proteins on different tissues.

### 2.5. LcPYL Genes Specifically Respond to Abiotic Stresses in L. hybrid

As one of the most important stress-induced phytohormones, ABA has been reported to control adaptive responses toward environmental stresses, such as drought and extreme temperature [29,30]. The *PYR/PYL* family genes act as the downstream factors of ABA signaling, mediating ABA perception and signal transduction [31]. To study the function of *LcPYL* genes, we analyzed the gene transcription pattern in *L. hybrid* under abiotic stresses. According to the reported transcriptome of *L. hybrid* leaves, the *LcPYL* genes were observed to differently respond to drought stress simulated with 15% PEG6000. The FPKM values showed that *Lchi11622* was highly upregulated under drought stress, *Lchi13641* was relatively upregulated, and *Lchi01385* and *Lchi00864* were not changed obviously; however, *Lchi16997* was downregulated (Figure 7A). To verify the expression pattern of drought-responsive genes, we tested the expression level of *Lchi11622* and *Lchi13641* in root, stem and leaf of *L. hybrid* treated with 20% PEG6000 by qPCR (Figure 7B,C). The results showed that *Lchi11622* was only induced by drought in leaves (not in root or stem), and its expression level in leaves under drought stress for 1 h and 3 days, was 6.98 and 12.04 folds of the gene expression in roots under normal condition, respectively (Figure 7B). The transcription of *Lchi13641* was found to be induced in leaves and stems under drought stress, but its expression level in roots was downregulated compared with the gene expression in stems under normal conditions (Figure 7C). The expression trends of genes in leaves tested by qPCR are in accordance with the transcriptome under drought treatment.

The transcriptome data of *L. hybrid* suffering from temperature stresses revealed that the *LcPYL*s reacted a different level compared with those under drought stress. The transcription of *Lchi11622* and *Lchi01385* was not obviously changed, and *Lchi00864* and *Lchi13641* were lightly downregulated; *Lchi16997* was the most responsive member and was upregulated under 37 °C for 1 h, 3 h and 3 days (Figure 8A). Accordingly, the qPCR data showed a similar expression pattern for *Lchi16997* (Figure 8B). However, low-temperature treatment at 4 °C revealed another expression pattern for *LcPYL* members (Figure 9A). The expression level of *Lchi11622*, *Lchi01385* and *Lchi16997* was upregulated after cold treatment, but they responded to cold stress at various treatment stages (Figure 9A). *Lchi16997* responded to cold when treated for 1 h and 3 h; however, *Lchi11622* and *Lchi01385* exhibited high transcription under cold treatment for 1 day and 3 days (Figure 9A). *Lchi00864* responded to cold 1 h treatment (Figure 9A). The qPCR data for *Lchi11622* and *Lchi01385* indicate that these genes positively responded to cold stress, with gene expression further improved by ABA treatment (Figure 9B–E). Compared with normal conditions, the expression levels of *Lchi11622* and *Lchi01385* in *L. hybrid* under low temperature for 3 days increased by 14 and 24 times, respectively, and slightly increased under low temperature and ABA treatment for three days compared with only low-temperature treatment. According to the results, we found that *Lchi11622* positively responded to cold stress but not hot stress, whereas *Lchi01385* positively responded to cold stress but negatively responded to hot treatment. Interestingly, *Lchi16997* was found to positively respond to both cold and hot treatment. These results suggest that *LcPYL*s positively responded to drought, hot and cold stresses but with specific expression patterns in response to various abiotic stresses.

**Figure 8 plants-12-02609-f008:**
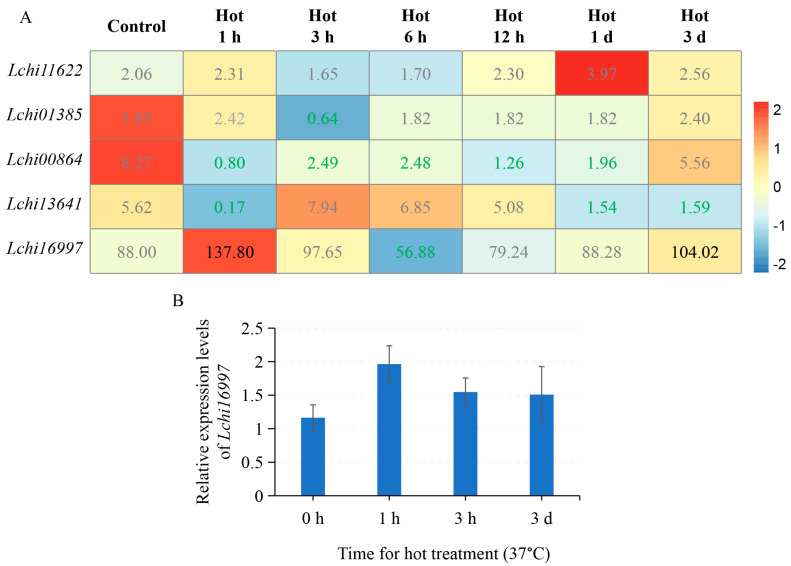
*LcPYL*s respond to heat stress. (**A**) The expression levels of *LcPYL*s are displayed as colored boxes with FPKMs from transcriptome data of *L. chinense* treated with 37 °C for the indicated times. Red represents a high expression level, and blue represents a low expression level. We marked the differentially expressed genes in the Figure 8A and Figure 9A with green FPKM values for downregulated genes and dark FPKM values for upregulated genes. (**B**) The relative expression levels of *Lchi16997* in plants treated with 37 °C for the indicated times were quantified by qPCR. h: hour; d: day.

**Figure 9 plants-12-02609-f009:**
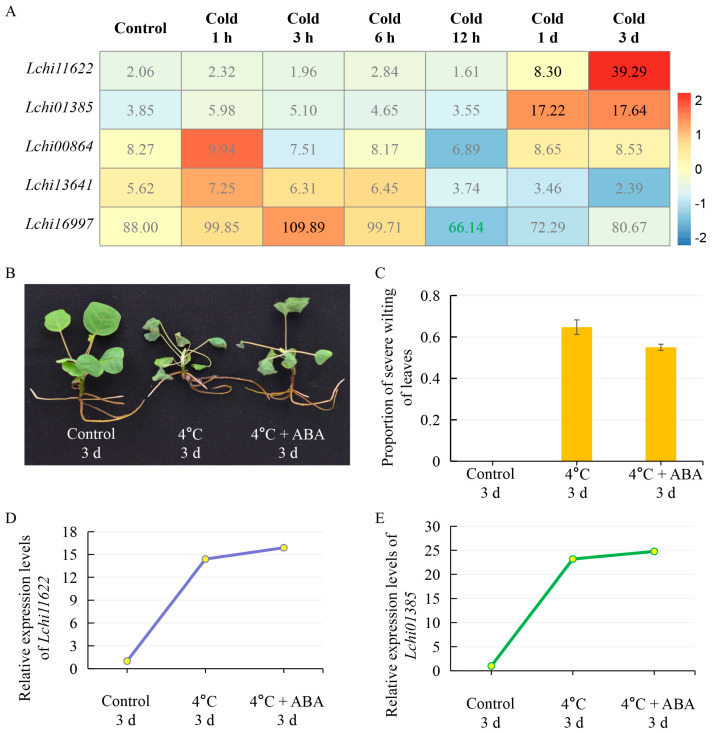
*LcPYL*s respond to cold stress. (**A**) The expression levels of *LcPYL*s are displayed as colored boxes with FPKMs from transcriptome data of *L. chinense* treated with 4 °C for the indicated times. Red represents a high expression level, and blue represents a low expression level. We marked the differentially expressed genes in the Figure 8A and Figure 9A with green FPKM values for downregulated genes and dark FPKM values for upregulated genes. (**B**) Phenotype of plants treated with 4 °C or 4 °C combined with 50 mg/L ABA for 3 d. (**C**) Proportion of plants with wilting leaves under the above treatments. (**D**,**E**) The relative expression levels of *Lchi11622* (**D**) and *Lchi01385* (**E**) in plants from (**B**) were quantified by qPCR. h: hour; d: day.

### 2.6. LcPYL Genes Specifically Respond to Biotic Stresses in L. hybrid

To explore the response of *LcPYL*s to biotic stresses, the *L. hybrid* leaves infected by scale insect and whitefly were harvested for qPCR tests. The leaves covered by the insects show disease spots that were dark green in the middle and yellow at the margin (Figure 10A–E). However, the *LcPYL*s were differentially transcribed in these leaves, as revealed by the results of the qPCR test. The relative expression levels of *Lchi13641* and *Lchi11622* in infected leaves increased by 1.76- and 3.75-fold relative to normal leaves, respectively. The relative expression levels of *Lchi00864* and *Lchi01385* in infected leaves were downregulated, while the transcription of *Lchi16997* was not changed obviously (Figure 10F). These results indicate that the *LcPYL*s from *L. hybrid* variously responded to biotic stresses caused by scale insect and whitefly.

### 2.7. Subcellular Localization of LcPYL Genes

To verify the subcellular localization of stress-responsive ABA receptors, Lchi11622, Lchi16997 and Lchi01385 were inserted into the *pJIT166-GFP* vector to overexpress LcPYL-GFP fusion proteins. As shown in Figure 11, the four fields from left to right are bright field, GFP field, H2B-mChrry field and merged field, and the empty GFP vector is used as the positive control. The results show that Lchi11622, Lchi16997 and Lchi01385 proteins were located in the cytoplasm and nucleus, which is consistent with the reported results of the localization of AtPYLs and OsPYLs family members in the cytoplasm and nucleus. This experiment shows that PYLs proteins are located in similar subcellular structures in different plant species, supporting their roles as ABA receptors.

## 3. Discussion

### 3.1. A Limit Number of LcPYL Proteins

PYL family proteins are critical factors for plant adaptation to abiotic and biotic stresses in the ABA signaling pathway, which have been identified from various plant species [32,33]. In *Arabidopsis*, 14 AtPYLs have been confirmed as ABA receptors, which were classified into three clades in the phylogenetic analysis [19]. In our study, only five PYL proteins were identified as ABA receptor members from the genome sequence of *L. chinense*, comprising 42% OsPYLs, 36% AtPYLs and 17% GhPYLs. Nevertheless, the phylogenetic study of LcPYLs and AtPYLs showed that Lchi01385, Lchi00864, AtPYR1 and AtPYL1-3 were classified into clade I; Lchi13641, Lchi11622, AtPYL4-6 and AtPYL11-13 were classified into clade II; and Lchi16997 and AtPYL7-10 were classified into clade III. These results suggest that although the LcPYL members are limited, the types of LcPYLs are complete compared with the classification of AtPYLs [11].

A previous phylogenetic study revealed the interesting correlations between the phylogeny of the MADS-box gene family and functional evolution of plants, which indicates that the genes falling into the same clade trend to function similarly. Therefore, the classification of LcPYLs and AtPYLs might suggest a limited number of LcPYL members but with complete function belonging to this gene family. Our results are consistent with the VvPYLs identification from *V. vinifera*, which revealed that only nine proteins were identified as PYL family members in the *V. vinifera* genome [20]; compared with other plant species, VvPYLs comprise 75% OsPYLs [21], 64% AtPYLs [19] and 33% GhPYLs [26].

*L. chinense*, as a perennial arbor, has shorter iteration times, resulting a lower probability of gene mutants affecting gene duplication than plant species with a short lifespan. For instance, *Arabidopsis* and rice complete their life in a few months, which leads to the generation of more reproductive processes than *L. chinense* [34,35]. New generations may evolve for a better adaptability to the environment, resulting in gene mutation and duplication. In a similar manner, *V. vinifera* is one kind of fruiter but also a perennial plant that has a much longer lifespan but fewer PYL members than *Arabidopsis* and rice. Therefore, we suppose that a limited number of *LcPYL* genes might be caused by the fewer iterations during evolutionary history.

### 3.2. LcPYLs Specifically Respond to Various Abiotic Stresses

In our study, five LcPYLs were identified from the genome of *L.chinense* that were found to specifically respond to different abiotic stresses (Figure 7, Figure 8 and Figure 9). Lchi11622 and Lchi13641 positively and significantly responded to drought stress simulated by 15% PEG6000; however, the transcription of other *LcPYL*s was not changed obviously under drought treatment (Figure 7A). Only the expression level of Lchi16997 was found to be upregulated under 37 °C treatment (Figure 8A). The expression levels of Lchi11622 and Lchi01385 were upregulated under cold stress, but the expression levels of the other three LcPYLs were not significantly changed (Figure 9A). These results demonstrate the specific expression of *LcPYL*s in response to various abiotic stresses. According to the analysis of cis-acting elements, we found that the promoters of *LcPYL*s contain various cis-acting elements in response to hormones, such as auxin, MeJA, salicylic acid and ABA, or abiotic stresses, such as low temperature and drought. These *LcPYL* promoters share some common cis-acting elements but also include their specific elements (Figure 5B). In our study, we found that all *LcPYL* promoter sequences contain ABA response element but at different levels. For instance, Lchi00864 has three ABRE elements and is identified as a cis-acting element involved in ABA responsiveness, Lchi01385 contains three ABA-responsive elements (one ABRE element and two AAGAA-motifs) and Lchi16997 has two AAGAA motifs. Lchi13641 and Lchi11622 contain one ABRE element and one AAGAA-motif but different numbers of other hormone or stress cis-elements. All these identities might be the reason for different spatiotemporal responses of *LcPYL* genes to various environmental stresses [36,37].

## 4. Materials and Methods

### 4.1. Plant Materials and Stress Treatment

In this study, *L. hybrid* seedlings were generated from callus-based somatic embryogenesis of *L. hybrid*. The *L. hybrid* seedlings were planted in pots with soil in a greenhouse at 22 °C under16 h light/8 h dark and 75% relative humidity for four weeks, then irrigated with 15% PEG6000 to simulate drought stress for three days. For temperature stress treatment, the prepared plants were moved into a chamber at 4 °C or 37 °C for three days. In order to study the response of *LcPYL* genes to ABA, the *L. hybrid* plants in a cold environment were sprayed with 50 mg/L ABA. The plants cultured under the normal condition were set as the control group. During the stress treatment, the *L. hybrid* plants were harvested at 1 h, 3 h and 3 d using liquid nitrogen and stored at −80 °C for RNA extraction. To check the transcription of *LcPYL*s in response to biotic stress, the first and second *L. hybrid* leaves infected by female whitefly and scale insect were harvested for qPCR testing. The plants grown under normal conditions were taken as the control. The samples were frozen in liquid nitrogen and stored at −80 °C for RNA extraction. Three biological replicates were conducted for each treatment.

### 4.2. Genome-Wide Identification of LcPYL Family Genes

The protein sequences of 14 AtPYLs were downloaded from the *Arabidopsis* information resource (TAIR 10, https://www.arabidopsis.org/, accessed on 7 August 2022) and employed as queries to search the *L. chinense* genome data using BLASTp [28]. The obtained proteins were then searched in the Pfam database (https://www.ebi.ac.uk/interpro/entry/pfam/#table, accessed on 7 August 2022) [38]. The Polyketide_cyc2 domain (PF10604) and Polyketide_cyc domain (PF03364) were found in the 13 harvested LcPYLs. The NCBI-CDD (https://www.ncbi.nlm.nih.gov/cdd/, accessed on 7 August 2022) database was further used to verify whether the candidate sequences have a PYR/PYL (RCAR)-like domain (cd07821) [39,40]. As a consequence, 6 PYLs were identified from the *L. chinense* genome. Based on the ABA receptor characteristics, 5 LcPYLs were finally confirmed by the conserved gate and latch motifs for ABA binding [16].

### 4.3. Analysis of Gene Exon–Intron Structures and Protein Conserved Motifs

*L. chinense* genome data and the CDS of 5 *LcPYL* genes were uploaded to TBtools (v1.108) for gene structure analysis [41]; the alignment of LcPYLs was conducted using Clustal Omega (v1.2.4) online software [42]. Prediction of 3D structures for LcPYL proteins was conducted using SWISS-MODEL online software (https://swissmodel.expasy.org/, accessed on 25 August 2022) and PyMOL (https://pymol.org/2/, accessed on 25 August 2022) [43,44]. The conserved motifs of 5 LcPYLs and 14 AtPYLs were analyzed by MEME online software (https://meme-suite.org/meme/doc/meme.html, accessed on 7 August 2022) with 10 as the maximum motif number setting and keeping default values for the remaining parameters [45,46].

### 4.4. Basic Information of LcPYL Proteins

The number of amino acids, molecular weight (MW), isoelectric point (pI), instability index, aliphatic index and grand average of hydropathicity of LcPYLs were studied with ExPASy online software (https://web.expasy.org/protparam/, accessed on 25 August 2022) [47]. The subcellular localization of LcPYL proteins was predicted by Wolf PSORT online software (https://wolfpsort.hgc.jp/, accessed on 25 August 2022) [48]. In addition, the secondary structures of LcPYL proteins were predicted by SOPMA using NPS online software (https://npsa-prabi.ibcp.fr/cgi-bin/npsa_automat.pl?page=/NPSA/npsa_server.html, accessed on 25 August 2022) [40].

### 4.5. Cis-Acting Elemental Analysis of LcPYL Gene Promoters

In order to explore the cis-acting elements existing in the promoter sequences of *LcPYL* genes, we used TBtools (v1.108) to extract 3000 bp upstream sequences of *LcPYL* open reading frames (ORFs) (including the 5′ untranslated region) from the *L. chinense* genome. Then, the obtained promoter sequences of 5 *LcPYL* genes were submitted to PlantCARE online software (http://bioinformatics.psb.ugent.be/webtools/plantcare/html/, accessed on 25 August 2022.) for cis-acting element prediction [49].

### 4.6. Phylogenetic Analysis

The sequence alignment and evolutionary study of 5 LcPYLs and 14 AtPYLs were performed using the maximum likelihood method in Mega 11 in with 1000 bootstrap replications and the JTT model. The AtPYL accession numbers and gene names are shown in Figure 3A. The phylogenetic tree of PYLs from *L. hybrid*, *A. thaliana*, *O. sativa* and *V. vinifera* was built using TBtools (v1.108) with maximum-likelihood estimation and 1000 bootstrap replications. Phylogenetic tree beautification was performed using iTOL online software (https://itol.embl.de/, accessed on 25 August 2022.) [50].

### 4.7. Quantitative Real-Time PCR Analyses

Quantitative real-time PCR (qPCR) analysis was performed to confirm the response of *LcPYL*s to drought, cold and hot treatment. The total RNA was extracted from root, stem and leaf tissue of *L. hybrid* plants treated with 15% PEG6000 at 4 °C and 37 °C, respectively. Total RNA extraction was conducted with a Eastep^®^ Super Total RNA Purification Kit (Promega, Shanghai, China), followed by genomic DNA digestion using DNase I prepared in the kit. The quality of total RNA was evaluated by ultraviolet spectrophotometry and gel electrophoresis. The cDNA was synthesized using a HiScript III 1st Strand cDNA Synthesis Kit (+gDNA wiper) (Vazyme Biotech, Nanjing, China) following the manufacturer’s instructions.

qPCR was performed using TB Green^®^ Premix Ex Taq™ (Takara, Dalian, China) on a LightCycler^®^480 qPCR detection system (Roche, Basel, Switzerland) following the manufacturer’s instructions. The expression level of *LcPYL* genes was normalized according to the expression level of 18S RNA in *L. hybrid* [51]. Three biological repeats and three experimental replicates were performed for the qPCR test. Specific primers of *LcPYL* genes and 18S RNA for qPCR are listed in Appendix A.

### 4.8. Gene Clone and Subcellular Localization Analysis

To verify the subcellular localization of stress-responsive *LcPYL*s predicted by the WoLF PSORT online tool, specific primers (listed in Appendix A) of Lchi01385, Lchi11622 and Lchi16997 were designed for PCR amplification using *L. hybrid* cDNA as templates to obtain their ORFs. The sizes of electrophoresis bands of PCR products were consistent with the expected sizes of the target genes. The target fragments obtained from gel extraction purification were inserted into the *pJIT166-GFP* vector to transiently express LcPYL-GFP fusion proteins under the control of the 2 × CaMV 35S promoter (35S). Then, each *p2 × 35S:LcPYL-GFP* vector and *p35S:H2B-mCherry* vector were simultaneously transformed into *L. hybrid* protoplasts, which were prepared based on the method proposed by Huo (2017) [52]. After three days, the images of green GFP and red H2B-mCherry fluorescence from the transformed protoplasts were observed and captured by a Zeiss LSM 800 laser scanning confocal microscope.

## 5. Conclusions

In this study, we identified five stress hormone ABA receptors from the *L. chinense* genome through sequence blast and characteristic confirmation that were further verified by sequence alignment and conserved motif analysis. The results revealed that these five *LcPYL*s contain a conserved gate and latch motif for ABA binding. The study of cis-acting elements existing in *LcPYL*s promoters indicates the potential response of these *LcPYL*s to various hormones and stresses. The transcriptome data of *L. hybrid* leaves and the results of qPCR test revealed the differential expression of *LcPYL* genes under different stresses, suggesting that these *LcPYL*s specifically respond to various stresses. The subcellular localizations of these stress-responsive LcPYLs were also analyzed in *L. hybrid* protoplasts. These results provide a foundation for functional exploration of *PYL*s from this elite tree species and support the understanding of the molecular mechanism of *L. hybrid* in coping with stresses. However, there are still some issues that need further verification and resolution. For example, the function of *LcPYL*s in response to various stresses could be explored by gene overexpression and knockout in *L. hybrid*. Additionally, the genetic redundancy between *LcPYL*s also needs to be clarified in this tree species.

## Figures and Tables

**Figure 1 plants-12-02609-f001:**
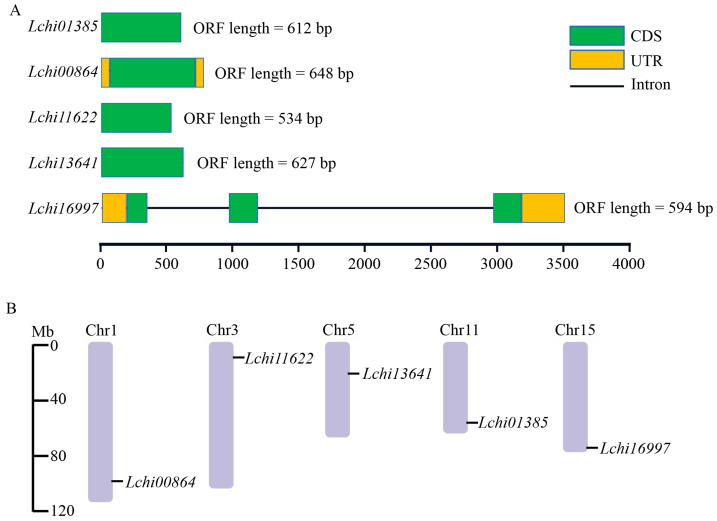
The gene architectures and distributions of *LcPYL*s on *L. chinense* chromosomes. (**A**) Architectures of *LcPYL*s including coding sequences (CDS) and the untranslated region (UTR). (**B**) The locations of *LcPYL*s on *L. chinense* chromosomes from top (start) to bottom (end) according to genome annotation. Chr: chromosome; numbers following ‘Chr’ are the chromosomal numbers.

**Figure 2 plants-12-02609-f002:**
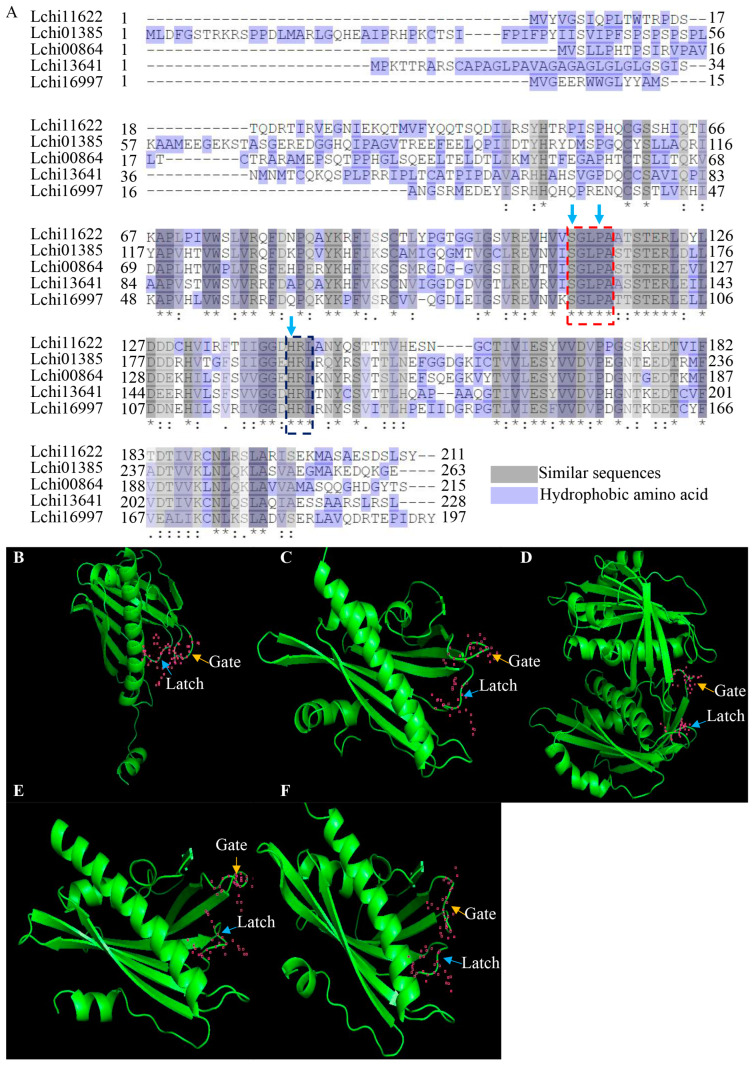
The sequence alignment and overall structures of LcPYL proteins. (**A**) Alignment of LcPYL protein sequences by Uniprot online software. The amino acids with grey shade are similar sequences; ‘*’ indicates that the amino acid residues are completely consistent; ‘:’ represents amino acid residues with particularly similar properties; ‘.’ represents amino acid residues with weakly similar properties. The amino acids with purple shade are hydrophobic. The red box indicates the conserved gate, and the black box indicates the latch residues. The key amino acids of gate and latch are noted with blue arrows. (**B**–**F**) Three-dimensional structures of Lchi01385 (**B**), Lchi00864 (**C**), Lchi11622 (**D**), Lchi13641 (**E**) and Lchi16997 (**F**) with conserved gate and latch motif for ABA binding were predicted using SWISS MODEL software. Orange and blue arrows represent the conserved gate and latch of ABA signal receptors, respectively. The pink dots indicate the skeleton of key amino acids of gates (SGLPA) and latches (HRL).

**Figure 3 plants-12-02609-f003:**
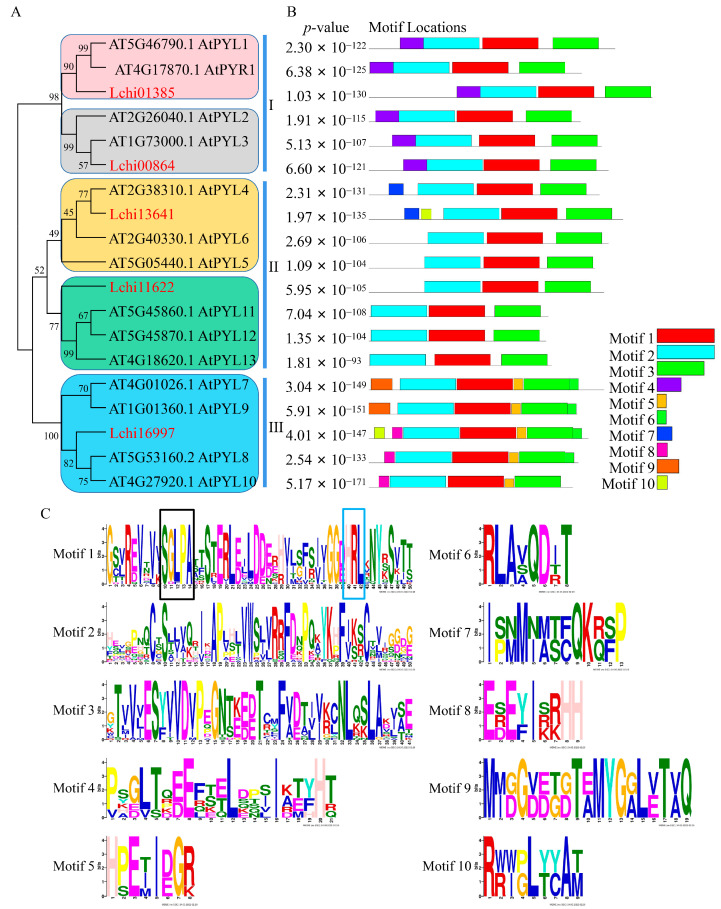
Phylogenetic study and conserved motif analysis for LcPYL proteins. (**A**) Phylogenetic relationship between 5 LcPYLs and 14 *Arabidopsis* PYLs. The phylogenetic tree was generated using the maximum likelihood method in Mega 5 with 1000 bootstraps. (**B**) Conserved motif distributions of PYL proteins. The 10 boxes with different colors on the right side indicate 10 conserved motifs. (**C**) The conservative sequences of 10 conserved motifs from (**B**). The black and blue boxes represent the conserved gate and latch of ABA signal receptors, respectively.

**Figure 4 plants-12-02609-f004:**
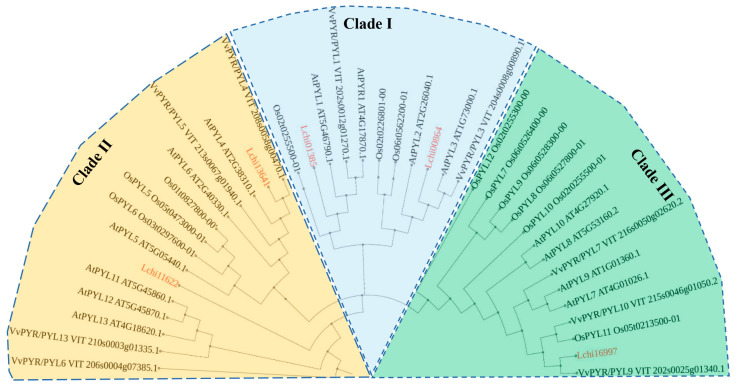
Phylogenetic study of *PYL*s from *L. chinense* and other plant species. The phylogenetic tree of *LcPYL* family genes and their homologous genes from *A. thaliana*, *O. sativa* and *V. viniferaother* was constructed using Mega 11 via the maximum likelihood method with 1000 bootstrap replications and the JTT model. Red gene IDs represent the genes from *L. chinense*.

**Figure 5 plants-12-02609-f005:**
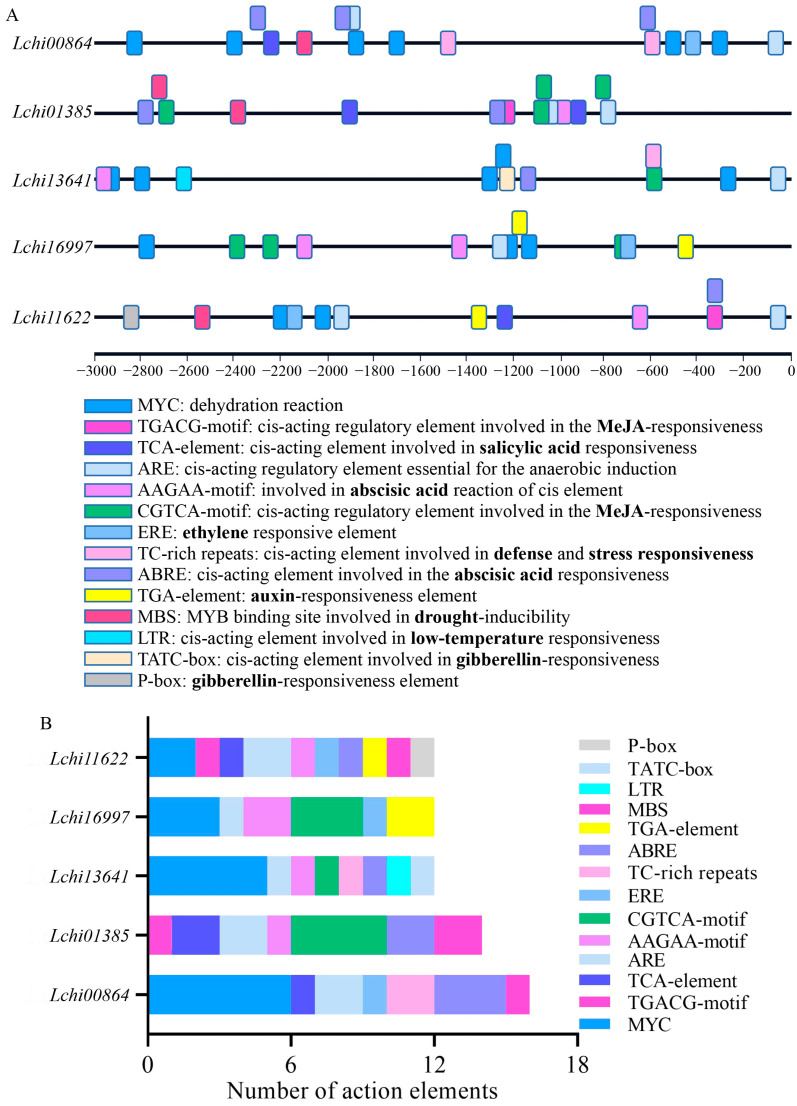
Cis-acting elements existing in the promoters of *LcPYL*s. (**A**) The distributions of cis-acting elements involved in various hormones and stresses are displayed with different color boxes. 3 kb upstream sequences of LcPYL ORFs including 5′UTR were extracted from the *L. chinense* genome for cis-acting elemental analysis. (**B**) Numbers of cis-acting elements from (**A**).

**Figure 6 plants-12-02609-f006:**
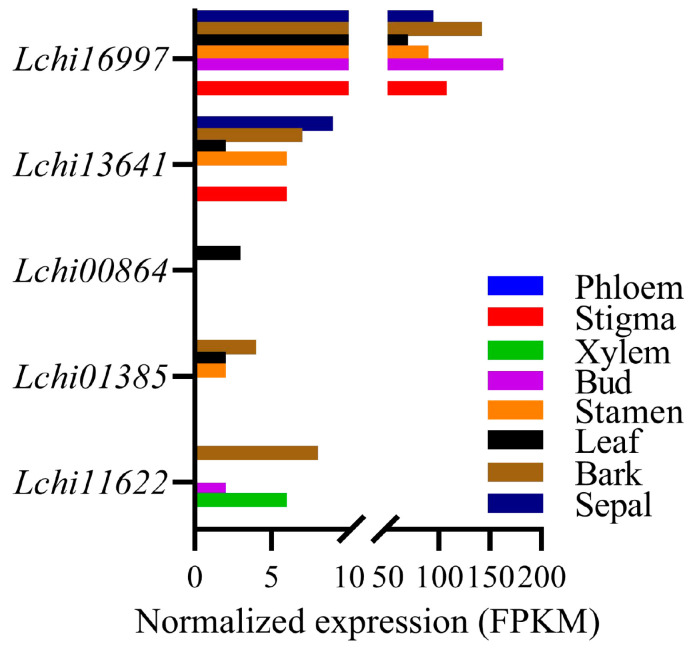
Expression level of *LcPYL* genes in different tissues of *L. chinense*. The transcriptome data of various *L. chinense* tissues including phloem, stigma, xylem, bud, stamen, leaf, bark and sepal reveal the expression levels of *LcPYL* genes. FPKM, fragments per kilobase million reads.

**Figure 7 plants-12-02609-f007:**
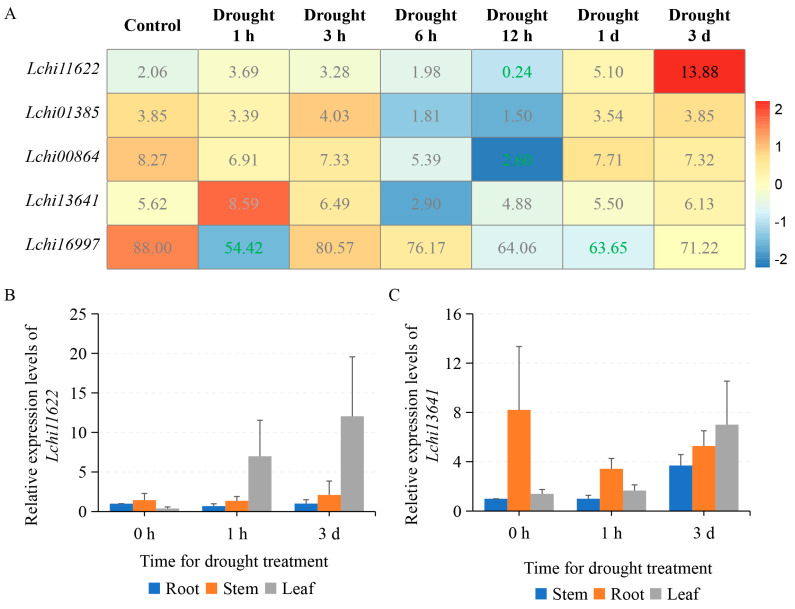
*LcPYL*s respond to drought stress. (**A**) The expression levels of *LcPYL*s are displayed as colored boxes with FPKMs from transcriptome data of *L. chinense* treated with drought stress simulated with 15% PEG6000. Red represents a high expression level, and blue represents a low expression level. We marked the differentially expressed genes in the Figure 8A and Figure 9A with green FPKM values for downregulated genes and dark FPKM values for upregulated genes. (**B**,**C**) The relative expression levels of *Lchi11622* (**B**) and *Lchi13641* (**C**) in root, stem and leaf treated with 20% PEG6000 for 0 h, 1 h and 3 d, respectively, were quantified with qPCR. h: hour; d: day.

**Figure 10 plants-12-02609-f010:**
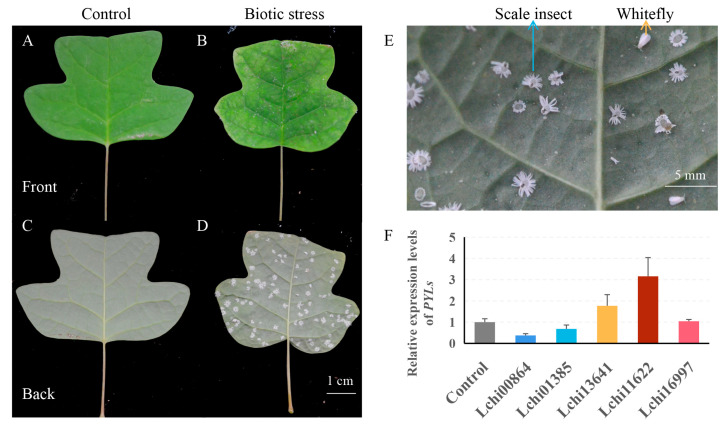
*LcPYL*s variously respond to biotic stresses. (**A**–**D**) Phenotype of *L. hybrid* leaves under normal conditions or infected with scale insect and whitefly. (**E**) Two types of insects (scale insect and whitefly) causing the biotic stresses in this study. (**F**) The relative expression levels of *PYL*s in *L. hybrid* leaves from (**A**–**D**) were tested by qPCR.

**Figure 11 plants-12-02609-f011:**
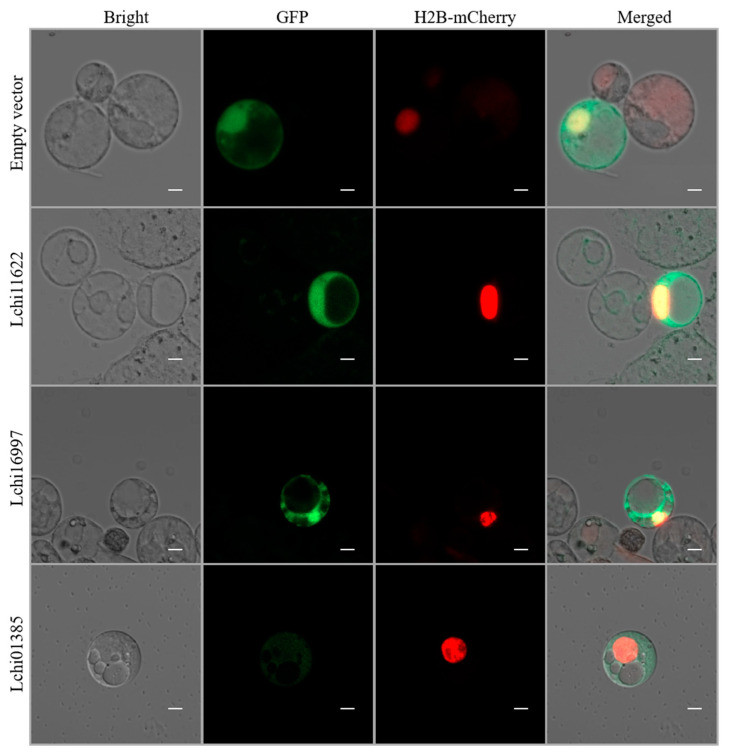
Subcellular localization of stress-responsive *LcPYL*s including Lchi11622, Lchi16997 and Lchi01385 in *L. chinense* protoplast. The cells overexpressing LcPYL-GFP fusion protein and H2B-mCherry (nuclear marker protein) are shown with fields as indicated. The vector only with GFP overexpression was taken as the positive control. Scale bar: 5 µm.

**Table 1 plants-12-02609-t001:** Characteristics of LcPYL proteins.

Gene ID	*Lchi01385*	*Lchi00864*	*Lchi11622*	*Lchi13641*	*Lchi16997*
aa ^1^	204	215	178	209	197
MW (kd) ^2^	22.66	23.75	19.75	22.55	22.53
pI ^3^	5	5.91	5.74	7.65	5.75
Instability index	38.31	33.28	60.21	48.32	47.14
Aliphatic index	78.28	86	85.9	90.43	90.41
GRAVY ^4^	−0.415	−0.226	−0.157	−0.056	−0.427
Subcellular location	Cytoplasm	Chloroplast	Chloroplast	Chloroplast	Cytoplasm

^1^ Amino acid; ^2^ molecular weight; ^3^ isoelectric point; ^4^ grand average of hydropathicity.

**Table 2 plants-12-02609-t002:** Secondary structure prediction of LcPYL proteins.

	Lchi01385	Lchi00864	Lchi11622	Lchi13641	Lchi16997
Alpha helix %	35.78 (73 aa ^1^)	37.67 (81 aa)	42.7 (76 aa)	32.06 (67 aa)	40.61 (80 aa)
Extended strand %	18.63 (38 aa)	19.07 (41 aa)	20.22 (36 aa)	19.14 (40 aa)	17.26 (34 aa)
Beta turn %	4.41 (9 aa)	6.05 (13 aa)	4.49 (8 aa)	4.78 (10 aa)	4.06 (8 aa)
Random coil %	41.18 (84 aa)	37.21 (80 aa)	32.58 (58 aa)	44.02 (92 aa)	38.07 (75 aa)

^1^ Amino acid.

## Data Availability

The abiotic stress transcriptome data of *L. hybrid* are annotated with accession number PRJNA679101 and can be downloaded through NCBI (https://www.ncbi.nlm.nih.gov/bioproject/PRJNA679101/, accessed on 7 August 2022). The complete genome, transcript/protein sequences and genome feature files of *Lchi* were downloaded from https://www.ncbi.nlm.nih.gov/assembly/GCA_003013855.2, accessed on 7 August 2022.

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
