# Peer review of "PYL Family Genes from Liriodendron chinense Positively Respond to Multiple Stresses"

_plants, 2023, doi:10.3390/plants12142609_

Round 1

Reviewer 1 Report

The study is in interest and helpful in the field of horticulture, but before acceptance authors should explain few more points in their study. Below are few comments to improve and broaden the study.

1.       As the study explain the response of plant after stress situations, I suggest that authors should also perform experiments for testing biotic stress response such as hormone level after insects and worms attack.

2.       Explain the detail of sources from where authors gather Information about latch motifs? And highlight the significance of latch receptors in few sentences.

3.       Provide the sequences used to build phylogenetic tree

4.       Based on the transcriptomic data provide the list of differentially expressed genes at different temperature range and during drought stress and explain their connection.

5.       Grammar and typos errors should be resolved before acceptance.

Grammar and typos errors should be resolved before acceptance.

Reviewer 2 Report

I have reviewed the manuscript entitled “PYL Family Genes From Liriodendron chinense Positively Re-spond To Multiple Abiotic Stresses” which was submitted for the possible publication in plants. My comments and suggestions are as follow:

The phytohormone abscisic acid (ABA) plays important roles in plant growth and stress responses.” This statement does not linked with the present title.

“Lchi01385 and Lchi16997 positively respond to cold and hot stress, respectively.” By which reference you have updated it?

Need to write future recommendation at the end of the abstract section.

“Higher plants have evolved a high plasticity for adaptation to environmental chal-lenges such as drought, cold and hot stresses, which severely influence plant normal de-velopment and productivity” The mentioned reference is too old and weak please add some latest reference such as:

Ali, M., Wang, X., Haroon, U., Chaudhary, H.J., Kamal, A., Ali, Q., Saleem, M.H., Usman, K., Alatawi, A., Ali, S., Hussain Munis, M.F., 2022. Antifungal activity of Zinc nitrate derived nano Zno fungicide synthesized from Trachyspermum ammi to control fruit rot disease of grapefruit. Ecotoxicology and Environmental Safety 233, 113311.

Gill, R.A., Ahmar, S., Ali, B., Saleem, M.H., Khan, M.U., Zhou, W., Liu, S., 2021. The Role of Membrane Transporters in Plant Growth and Development, and Abiotic Stress Tolerance. International Journal of Molecular Sciences 22, 12792.

Rasheed, A., Khan, A.A., Nawaz, M., Mahmood, A., Arif, U., Hassan, M.U., Iqbal, J., Saleem, M.H., Ali, B., Fahad, S., 2023. Development of Aluminium (Al)-Tolerant Soybean Using Molecular Tools: Limitations and Future Directions. Journal of Plant Growth Regulation.

“Based on the classification of AtPYLs (8), 40 PYL proteins from four plant species were also classified into three clades, designated clade I, clade II and clade III” Please revise it.

“Nevertheless, the clustering analysis of LcPYLs and AtPYLs showed that two LcPYLs were classified into clade I, another two LcPYLs were classified into clade II, the left one LcPYL was classified into clade III.” Please revise it.

Add some more future recommendations at the end of the conclusion.

Minor editing of English language required

Reviewer 3 Report

The paper “PYL Family Genes From Liriodendron chinense Positively Respond To Multiple Abiotic Stresses” by Xinru Wu, Junjie Zhu, Xinying Chen, Jiaji Zhang, Lu Lu, Zhaodong Hao, Jisen Shi and Jinhui Chen describes a stress response study of Liriodendron chinense, endangered endemic tree native to Central and South China a part of Northern Vietnam. L. chinense is grown as ornamental tree in UK, Ireland and USA, but not for example in Central Europe due to insufficient endurance to winter freeze.

Phylogenetic study, cis-acting element exploration, gene expressions using different types of tissue, drought, warm and cold temperature stress studies a specific expressions obtained according to individual stress factors are given in the manuscript. Only one level of drought stress (15% PEG) and relatively mild temperature conditions (37ºC, 4ºC) were studied, but the results clearly indicate, which of LcPYL genes takes part in the regulation under each particular type of stress.

The topic is both interesting – when the world finally admitted that current climate change is far more complicated system then simple “global warming”, adaptation to changing conditions are being studied on a wide scale to find best means of help for all affected organisms (except proliferation of pathogens, where environmentally friendly means for their limitation are investigated) – and important: endangered species like L. chinense are often hit the most, as climate change further diminishes their isolated endemic biotops without possibility for migration to more favourable conditions. The paper will be beneficial for the readers of the Plants journal.

The manuscript is well organised, mostly given in good, simple scientific English, results are presented clearly, with appropriate number of tables, charts and figures, and the results support the hypothesis.

From the formal point of view, there are some minor flaws the authors would like to check. Final control of the manuscript was, very probably, outsourced to a language agency that does not posses expert knowledge of the manuscript topic. Thus, L. chinense is called “precious” in the introduction, which is probably result of “rare, endemic endangered specie” being too long and replaced. These beautiful trees are, no doubt, precious to many, but the term creates the picture of a tree planted by a grand-grandfather and loved by the whole family, or even the “Lord of the Rings”-type “preciousness”. In Materials and methods, “...seedlings were planted in pots ... for four weeks, that were irrigated.” According to that sentence, it seems that the weeks were irrigated instead of pots or seedlings. Description of Fig. 7C says “reletive expressions”, because they use grammar software and they cannot check figures.

Thus, the paper will be recommended for minor revision, so the authors could fix some mistypes and re-phrase the text in several places. However, the manuscript bring important new information, it broadens the knowledge on plant stress regulation, and is recommended for acceptation.

As told before. Mostly, the English si fine. Final corrections were given to some non-expert externist, probably a language agency, and they only did what can be found by a computer check.

The authors deserve better. Please, let them check the manuscript once more and fix some minor issues.

Round 2

Reviewer 2 Report

Authors have made all the necessary changes; therefore paper should be accepted in the current form.

Thanks

Authors have made all the necessary changes; therefore paper should be accepted in the current form.

Thanks